

# Genetic investigation of population structure in Atlantic chub mackerel, *Scomber colias* Gmelin, 1789 along the West African coast

Salah eddine Sbiba[1,2], María Quintela[3], Johanne Øyro[3], Geir Dahle[3], Alba Jurado-Ruzafa[4], Kashona Iita[5], Nikolaos Nikolioudakis[3], Hocein Bazairi[1,6] and Malika Chlaida[2]

[1] Biodiversity, Ecology and Genome Laboratory, Faculty of Sciences, Mohammed V University in Rabat, Rabat, Morocco
[2] Research and Development Unit on Marine Biology, National Institute of Fisheries Research, Casablanca, Morocco
[3] Department of Population Genetics, Institute of Marine Research, Bergen, Norway
[4] Oceanographic Centre of the Canary Islands, Spanish Institute of Oceanography (IEO-CSIC), Tenerife, Spain
[5] National Marine Information and Research Centre (NATMIRC), Ministry of Fisheries and Marine Resources, Swakopmund, Namibia
[6] University of Gibraltar, Europa Point Campus, Natural Sciences and Environment Research Hub, Gibraltar, Gibraltar

Corresponding author
Malika Chlaida,
ma_chlaida@hotmail.com

## ABSTRACT

Sustainable management of transboundary fish stocks hinges on accurate delineation of population structure. Genetic analysis offers a powerful tool to identify potential subpopulations within a seemingly homogenous stock, facilitating the development of effective, coordinated management strategies across international borders. Along the West African coast, the Atlantic chub mackerel (*Scomber colias*) is a commercially important and ecologically significant species, yet little is known about its genetic population structure and connectivity. Currently, the stock is managed as a single unit in West African waters despite new research suggesting morphological and adaptive differences. Here, eight microsatellite loci were genotyped on 1,169 individuals distributed across 33 sampling sites from Morocco (27.39°N) to Namibia (22.21°S). Bayesian clustering analysis depicts one homogeneous population across the studied area with null overall differentiation ($F_{ST} = 0.0001^{ns}$), which suggests panmixia and aligns with the migratory potential of this species. This finding has significant implications for the effective conservation and management of *S. colias* within a wide scope of its distribution across West African waters from the South of Morocco to the North-Centre of Namibia and underscores the need for increased regional cooperation in fisheries management and conservation.

## INTRODUCTION

Sustainable stock management and conservation of marine resources require knowledge of the species' population variability in a bio-geographical context. Failure to take into consideration the underlying components of the population structure of a species (*e.g.*, spatio-temporal mixing of populations) can result in differential exploitation patterns and discrepancies between management measures among countries, which can lead to overexploitation of unique spawning components and subsequent resources declines (*e.g.*, *Allendorf et al., 2008*; *Kerr et al., 2017*). As such, a reliable delineation of the biological units is needed to maintain the sustainability of fisheries, especially as climate change and other human pressures on marine ecosystems continue to intensify (*Gissi et al., 2021*; *IPCC, 2022*; *Ramírez et al., 2022*).

Genetic and genomic tools are commonly used to accurately define stock structure of fish (*Ward, 2000*) and to assess connectivity among marine populations (*Bekkevold et al., 2015*; *Besnier et al., 2014*; *Dahle et al., 2018b*; *Hemmer-Hansen et al., 2019*; *Henriques et al., 2017*; *Le Moan, Bekkevold & Hemmer-Hansen, 2021*). Using genetic and genomic tools to identify stock structure has proved useful for a variety of issues such as updating management plans (*Mullins et al., 2018*; *Quintela et al., 2020*; *Saha et al., 2017*; *Westgaard et al., 2017*), cost-effective fisheries management enforcement (*Glover, 2010*; *Martinsohn et al., 2019*), "real-time" regulation of harvest (*Dahle et al., 2018a*; *Johansen et al., 2018*) and harmonizing biological and management units (*Aguirre-Sarabia et al., 2021*; *Leone et al., 2019*; *Rodríguez-Ezpeleta et al., 2019*), which is particularly important for transboundary stocks of highly migratory and commercial important fish. Microsatellites are one of such tool, which consist of short repeating sequences of DNA found in organisms (*O'Connell & Wright, 1997*; *Ellegren, 2004*). They are highly variable due to their high mutation rate and are ideal for genetic studies to assess genetic diversity, identify population structure, and conduct parentage analysis (*Ramya & Behera, 2023*).

Widely distributed species with shared/transboundary distributions (*Munro, Van Houtte & Willmann, 2004*) such as the Atlantic chub mackerel, *Scomber colias* Gmelin, 1789, are particularly challenging to manage due to the complexities of their stock structure, but make them a well-suited candidate for genetic analysis. *S. colias*, a medium-size pelagic schooling species belonging to the Scombridae family, inhabits the continental shelf down to 300 m depth throughout the Atlantic Ocean, the Mediterranean and the Black Seas (*Collette, 1986*; *Collette & Nauen, 1983*; *Scoles, Graves & Collette, 1988*). In the Northeast Atlantic, the species is distributed from southern European waters (Iberian Peninsula) southwards along the African coast to South of Africa, and is also found in the connected Mediterranean and Black Seas (*Castro Hernández & Santana Ortega, 2000*). The species is economically valuable and targeted by industrial and artisanal fisheries throughout its geographical distribution (*Castro Hernández & Santana Ortega, 2000*).

Various studies suggested significant differences between Atlantic chub mackerel from the Western and the Eastern Atlantic, supporting the existence of two different

populations occurring on either side of the Atlantic Ocean (*Costa et al., 2011*; *Roldán et al., 2000*; *Scoles, Graves & Collette, 1988*). In the NE Atlantic and surrounding waters, the available genetic evidence suggests significant lack of genetic diversity in *S. colias* indicating the presence of a large panmictic unit (*Rodríguez-Ezpeleta et al., 2016*). According to *Zardoya et al. (2004)*, the 5′ end of the mitochondrial control region indicates that *S. colias* behaves as a panmictic population in the Mediterranean Sea and the south of Portugal. Based upon microsatellites, *Stroganov et al. (2023)* found that in the southern region of Morocco (23°N–21°N), samples show a high genetic homogeneity, suggesting large degree of connectivity. Congeneric species such as *S. scombrus* reveal population differentiation only at a transatlantic scale (*Nesbø et al., 2000*), whereas *S. japonicus* displays weak genetic structure across the NW Pacific (*Cheng et al., 2015*), with local genetic differentiation seemingly linked to differences in spawning time and migratory behaviour (*Zeng, Cheng & Chen, 2012*).

Migration patterns could be potentially impacting population structure. Thus, in both African and European waters, the species would migrate from the wintering areas (mainly located off Mauritania, South Portugal and the inner part of the Bay of Biscay) towards northern areas in summer and, in the case of the Bay of Biscay, also towards the western Iberian Peninsula (*ICES, 2021*). Recent studies investigating the spatial variability of life-history parameters of *S. colias* suggested the existence of latitudinal trends with a mixing zone between Atlantic African, Mediterranean and Atlantic Iberian population components (*e.g.*, *Domínguez-Petit et al., 2022*). In the most recent study using otolith shape analysis, *Sbiba et al. (2024)* pointed out the existence of two populations, a northern one in Morocco from Larache (36°N) to Tarfaya (28°N), and a southern one from Tarfaya to Senegal (14°N). Thus, two stocks have been outlined according to biological traits: a northern one ranging from North Morocco to Cape Bojador in South Morocco (26°07′37″N, 14°29′57″W), and a southern one between Cape Bojador and the South of Senegal (https://firms.fao.org/firms/resource/10100/en). However, the lack of both robust evidence of stock identity (*ICES, 2021*) as well as information on population connectivity led to a joint assessment of both stocks in this area since 2003, thus highlighting the need for stock identity information in the NE Atlantic part of the species range (*FAO, 2020*). Morphometric differences on *S. colias* were also detected across five sampling sites in Morocco (*Bouzzammit & El Ouizgani, 2019*) as well as across three different locations in Tunisia (*Allaya et al., 2016*). Moreover, *Erguden et al. (2009)* used morphometric and meristic analyses to distinguish stocks of *S. colias* in seven locations throughout the Black, Marmara, Aegean, and Northeastern Mediterranean Seas, revealing the existence of two distinct groups. Overall, these studies also highlight the high potential of phenotypic plasticity of this species.

The migratory behaviour of *S. colias*, along with the high dispersal potential of its early life stages (eggs and larvae) within the open Atlantic Ocean is expected to favour connectivity and act as a homogenization factor. In addition, the West African coast is affected by several oceanic currents (see Fig. 1), *i.e.*, the Canary, Guinea, and Benguela Currents that can influence both dispersal and migration patterns, thus amplifying the
homogenization effects (*White et al., 2010*). Consequently, a pattern of genetic panmixia is foreseen on *S. colias* along the west coast of Africa. To verify this hypothesis, eight microsatellites were used to investigate the population structure of this species sampled from south of Morocco (~28°N) to Namibia (~22°S) at 33 locations.

# MATERIALS AND METHODS

## Sampling

Fish were collected during the scientific surveys on board the R/V Dr. Fridtjof Nansen carried out in 2017 and 2019 in the area stretching from Morocco (~28°N) to Namibia (~22°S) (Fig. 1). Sampling permits were obtained through the "Application of the Ecosystem Approach to Fisheries management considering climate and pollution impacts" (GCP/GLO/690/NOR). Pelagic trawl hauls were performed after echosounders target identification of fish schools and, when possible, 30 individuals were collected per trawling station (Table 1). The sampling procedure reflected the species' distribution (*Baird, 1977*; *Scoles, Graves & Collette, 1988*), whose southern limit of highest abundance seems to be placed at low latitudes in the southern hemisphere. Thus, some 75.5% of the individuals were collected north to Ghana whereas the southernmost countries such as Angola and Namibia only accounted for 7.5% of the individuals in line with the species biomass levels observed in those zones. Fin clips from 1,848 fish were collected and preserved in ethanol prior to DNA extraction.

## DNA isolation and genotyping

DNA was extracted using the Qiagen DNeasy 96 Blood & Tissue Kit in (Qiagen, Hilden, Germany) 96-well plates; each of which contained two or more negative controls. DNA concentration was quantified using a NanoDrop 8000 (Thermo Fisher Scientific, Waltham, MA, USA). The molecular markers used in the current study were initially developed for both *S. colias* (one) and *S. japonicus* (seven) (*Catanese et al., 2010*; *Chen et al., 2017*; *Zeng & Cheng, 2012*) as microsatellite cross-species amplification has proven a successful tool for fisheries management and conservation (*e.g.*, *Maduna et al., 2014*) formerly applied to *Scomber* species (*Tang et al., 2009*; *Zeng, Cheng & Chen, 2012*). Eight microsatellite loci isolated from *S. colias* (Sco2_1) and *S. japonicus* (SJ78, SJNT19, SJT5, SJT53, SJT122, SJT182, SJT199), respectively (Table S1 in Supporting Information), were genotyped in three multiplexed reactions. PCR amplifications were performed in a final volume of 10 µL containing approximately 20 ng DNA template, $1 \times$ buffer, 2–3 mM $MgCl_2$, 0.2 mM dNTPs, 0.2–0.5 µM of each primer and 1U GoTaq polymerase in an Applied Biosystems Gene Amp PCR Systems 2700 thermal cycler. PCR profiles consisted of 15 min denaturation at 95 °C followed by 27–32 cycles of 30 s denaturation at 94 °C, 90 s at annealing temperature (Table S1 in Supporting Information) and 90 s extension at 72 °C with a final step of 7 min at 72 °C. PCR products were analysed on an ABI 3730 Genetic Analyser and the 500LIZ$^{TM}$ size standard (Applied Biosystems, Waltham, MA, USA) was used to accurately determine the size of the fragments and allelic variation. Scoring was performed using the software Fragment Profiler 1.2 (Amersham Biosciences,

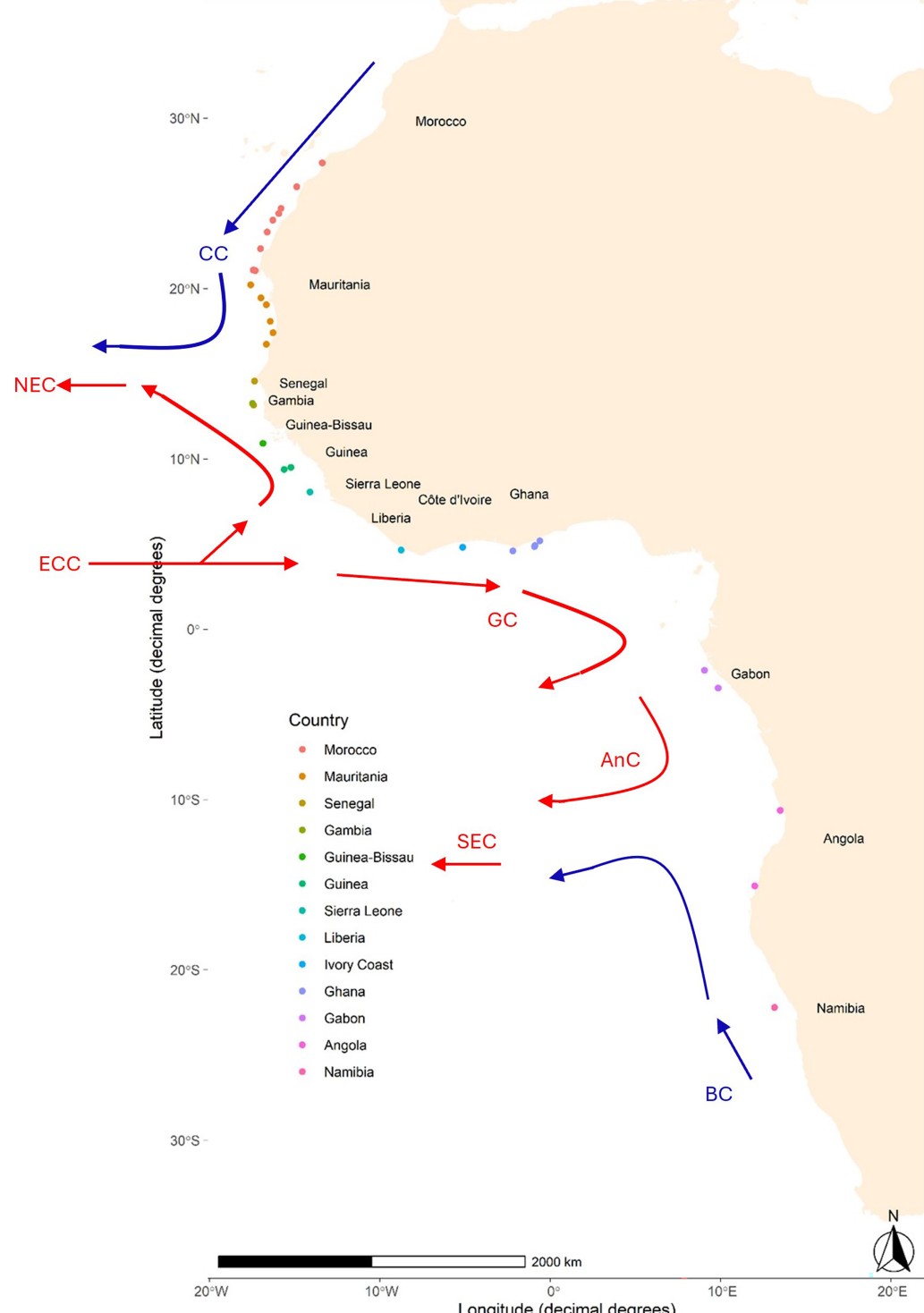

**Figure 1 Map of sampling locations of *Scomber colias* in the West coast of Africa.** Detailed information per sample can be found in Table 1. The arrows, based upon Fig. 1 in *Reid et al. (2016)*, indicate some of the major currents affecting the area and their relative temperature: Canary Current (CC), North Equatorial Current (NEC), Equatorial Counter Current (ECC), Guinea Current (GC), Angolan Current (AnC), and South Equatorial Current (SEC).

**Table 1 Summary statistics of *Scomber colias* genotyped at eight microsatellites.**

| Country | Sample | Latitude | Longitude | N | No alleles | Ar ($N >$ 20) | No private alleles | $H_o$ | u $H_e$ | $F_{IS}$ | Dev HWE (FDR) | Dev LD (FDR) |
|---|---|---|---|---|---|---|---|---|---|---|---|---|
| Morocco | MOR_1 | 27.39 | −13.38 | 51 | 128 | 12.1 | 0 | 0.791 ± 0.023 | 0.879 ± 0.013 | 0.090 ± 0.028 | 3 (2) | 2 (0) |
| | MOR_2 | 25.98 | −14.87 | 58 | 135 | 12.4 | 2 | 0.803 ± 0.033 | 0.881 ± 0.015 | 0.081 ± 0.032 | 1 (1) | 3 (1) |
| | MOR_3 | 24.71 | −15.8 | 45 | 130 | 12.6 | 0 | 0.780 ± 0.040 | 0.885 ± 0.011 | 0.108 ± 0.047 | 5 (4) | 6 (2) |
| | MOR_4 | 24.4 | −15.93 | 41 | 126 | 12.5 | 0 | 0.832 ± 0.028 | 0.885 ± 0.011 | 0.048 ± 0.029 | 5 (3) | 7 (1) |
| | MOR_5 | 24.02 | −16.28 | 44 | 117 | 11.8 | 0 | 0.763 ± 0.041 | 0.878 ± 0.014 | 0.122 ± 0.045 | 1 (1) | 2 (0) |
| | MOR_6 | 23.32 | −16.6 | 40 | 126 | 12.4 | 1 | 0.794 ± 0.037 | 0.873 ± 0.016 | 0.079 ± 0.036 | 3 (3) | 0 (0) |
| | MOR_7 | 22.35 | −17 | 51 | 126 | 12.4 | 1 | 0.823 ± 0.039 | 0.889 ± 0.009 | 0.064 ± 0.043 | 2 (1) | 10 (3) |
| | MOR_8 | 21.06 | −17.31 | 38 | 121 | 12.9 | 0 | 0.841 ± 0.031 | 0.891 ± 0.012 | 0.043 ± 0.032 | 4 (4) | 3 (0) |
| | MOR_9 | 21.09 | −17.42 | 39 | 125 | 12.7 | 1 | 0.811 ± 0.051 | 0.884 ± 0.013 | 0.072 ± 0.054 | 2 (2) | 1 (0) |
| Mauritania | MAU_1 | 20.22 | −17.58 | 41 | 124 | 12.5 | 0 | 0.803 ± 0.037 | 0.882 ± 0.013 | 0.078 ± 0.039 | 3 (1) | 3 (2) |
| | MAU_2 | 19.46 | −16.98 | 41 | 122 | 12.4 | 0 | 0.843 ± 0.041 | 0.884 ± 0.012 | 0.034 ± 0.043 | 4 (3) | 0 (0) |
| | MAU_3 | 19.05 | −16.66 | 34 | 118 | 12.8 | 0 | 0.777 ± 0.055 | 0.881 ± 0.014 | 0.106 ± 0.056 | 2 (2) | 1 (0) |
| | MAU_4 | 18.08 | −16.43 | 40 | 123 | 12.5 | 1 | 0.839 ± 0.037 | 0.879 ± 0.012 | 0.035 ± 0.034 | 1 (1) | 0 (0) |
| | MAU_5 | 16.73 | −16.66 | 44 | 122 | 11.9 | 0 | 0.812 ± 0.036 | 0.872 ± 0.016 | 0.059 ± 0.036 | 1 (0) | 0 (0) |
| | MAU_6 | 17.41 | −16.27 | 23 | 101 | 12.1 | 0 | 0.790 ± 0.041 | 0.866 ± 0.025 | 0.062 ± 0.052 | 2 (0) | 1 (0) |
| Senegal | SEN_1 | 14.56 | −17.35 | 44 | 129 | 12.2 | 4 | 0.778 ± 0.035 | 0.869 ± 0.015 | 0.095 ± 0.033 | 2 (1) | 1 (0) |
| | SEN_2 | 13.16 | −17.41 | 40 | 124 | 12.5 | 0 | 0.831 ± 0.023 | 0.882 ± 0.013 | 0.045 ± 0.024 | 3 (2) | 2 (0) |
| Gambia | GAM | 13.25 | −17.47 | 22 | 102 | 12.4 | 1 | 0.806 ± 0.052 | 0.877 ± 0.022 | 0.061 ± 0.052 | 2 (2) | 1 (1) |
| Guinea-Bissau | GUIBIS | 10.92 | −16.86 | 5 | 46 | n.a. | 0 | 0.725 ± 0.053 | 0.850 ± 0.023 | 0.034 ± 0.087 | 0 (0) | 0 (0) |
| Guinea | GUI_1 | 9.38 | −15.61 | 29 | 106 | 11.8 | 0 | 0.773 ± 0.038 | 0.869 ± 0.021 | 0.095 ± 0.036 | 3 (3) | 1 (0) |
| | GUI_2 | 9.51 | −15.22 | 24 | 102 | 12.1 | 0 | 0.802 ± 0.037 | 0.871 ± 0.014 | 0.058 ± 0.043 | 2 (1) | 0 (0) |
| Sierra Leone | SIER | 8.06 | −14.09 | 11 | 71 | n.a. | 1 | 0.839 ± 0.054 | 0.866 ± 0.019 | −0.011 ± 0.054 | 0 (0) | 0 (0) |
| Liberia | LIB | 4.66 | −8.74 | 33 | 116 | 12.7 | 0 | 0.814 ± 0.048 | 0.894 ± 0.010 | 0.075 ± 0.053 | 1 (1) | 1 (0) |
| Ivory Coast | IVCO | 4.81 | −5.13 | 23 | 96 | 11.5 | 1 | 0.828 ± 0.027 | 0.877 ± 0.014 | 0.034 ± 0.029 | 3 (2) | 3 (0) |
| Ghana | GHA_1 | 5.19 | −0.6 | 51 | 142 | 13.0 | 1 | 0.820 ± 0.025 | 0.886 ± 0.013 | 0.065 ± 0.025 | 2 (1) | 2 (0) |
| | GHA_2 | 4.59 | −2.19 | 43 | 124 | 12.3 | 0 | 0.806 ± 0.034 | 0.883 ± 0.012 | 0.077 ± 0.036 | 2 (1) | 0 (0) |
| | GHA_3 | 4.92 | −0.89 | 30 | 116 | 12.9 | 2 | 0.806 ± 0.032 | 0.881 ± 0.015 | 0.070 ± 0.032 | 2 (1) | 0 (0) |
| | GHA_4 | 4.86 | −0.92 | 35 | 114 | 11.9 | 0 | 0.798 ± 0.041 | 0.877 ± 0.015 | 0.076 ± 0.047 | 0 (0) | 0 (0) |
| Gambia | GAB_1 | −2.41 | 9.06 | 38 | 137 | 13.6 | 1 | 0.819 ± 0.023 | 0.884 ± 0.013 | 0.061 ± 0.019 | 0 (0) | 1 (0) |
| | GAB_2 | −3.45 | 9.87 | 24 | 104 | 12.2 | 0 | 0.813 ± 0.036 | 0.879 ± 0.014 | 0.055 ± 0.042 | 2 (1) | 1 (0) |
| Angola | ANG_1 | −10.64 | 13.52 | 27 | 111 | 12.5 | 0 | 0.792 ± 0.041 | 0.875 ± 0.019 | 0.077 ± 0.047 | 2 (1) | 0 (0) |
| | ANG_2 | −15.06 | 12.02 | 38 | 114 | 12.2 | 0 | 0.768 ± 0.057 | 0.873 ± 0.016 | 0.109 ± 0.061 | 1 (1) | 0 (0) |
| Namibia | NAM | −22.21 | 13.17 | 22 | 96 | 11.6 | 0 | 0.733 ± 0.049 | 0.861 ± 0.019 | 0.129 ± 0.051 | 1 (0) | 0 (0) |

**Note:**
Sampling sites per country, geographic coordinates (decimal degrees), number of individuals (N), total number of alleles, allelic richness based on a minimum sample of 20 diploid individuals, number of private alleles, observed heterozygosity $H_o$ (mean ± SE), unbiased expected heterozygosity u$H_E$ (mean ± SE), inbreeding coefficient $F_{IS}$ (mean ± SE), number of deviations from Hardy-Weinberg equilibrium (HWE) at α = 0.05, number of deviations from Linkage Disequilibrium (LD) at α = 0.05 with False Discovery Rate (FDR) correction.

Buckinghamshire, England). Automatically binned alleles were manually checked by two researchers prior to exporting data for statistical analyses.

As it was not always possible to obtain a minimum of thirty good-quality genotyped individuals per site, nearby sampling sites within countries were merged to achieve significant sampling sizes (*Hale, Burg & Steeves, 2012*). Sampling sizes ranged from 22 individuals (GAM in Gambia) to 58 (MOR_2 in Morocco). However, in Sierra Leone ($N = 11$), and Guinea-Bissau ($N = 5$) (Table 1), sampling sizes were too low and therefore these samples were excluded from some of the analyses.

## Statistical analysis

Loci were screened for null alleles, large allele dropouts and potential scoring errors due to stuttering bands using the software Micro-Checker v.2.2.3 (*Van Oosterhout et al., 2004*). The frequency of null allele(s) was estimated with the maximum likelihood method using the EM algorithm of *Dempster, Laird & Rubin (1977)* implemented in the software Genepop 7 (*Rousset, 2008*). To assess the effect of including loci with possible null allele(s) on population differentiation estimates, the software FreeNA (*Chapuis & Estoup, 2006*) was used to calculated both uncorrected and corrected $F_{ST}$ values. Confidence intervals (95%) of null frequencies were based on 1,000 bootstraps.

The statistical power of this microsatellites set to detect genetic differentiation was estimated using POWSIM 4.1 (*Ryman & Palm, 2006*), which uses $\chi^2$ and Fisher tests to assess whether the observed data set carries enough statistical power to detect a Nei's $F_{ST}$ value significantly larger than zero. The analyses were conducted maintaining effective population size constant ($N_e = 15,000$) following *Henriques et al. (2017)*, and changing the number of generations of drift since divergence (t: 15, 35, 100, 160, 200, 350 and 500 generations) to model different levels of $F_{ST}$ (0.0005, 0.0012, 0.0033, 0.0053, 0.0066, 0.0117, and 0.0165, respectively). The probability of rejecting the null hypotheses of no population differentiation (type I error) was also estimated ($F_{ST} = 0$, t = 0). Statistical power was determined as the proportion of tests indicating statistical significance ($p < 0.05$), with 1,000 replicates. In addition, to assess if this suite of twelve microsatellites would accurately discriminate between individuals in a population, the genotype accumulation curve was built using the function *genotype curve* in the R (*R Core Team, 2020*) package *poppr* (*Kamvar, Tabima & Grünwald, 2014*) by randomly sampling x loci without replacement and counting the number of observed multilocus genotypes (MLGs).

The total number of alleles, number of private alleles, and allelic richness per sample was calculated using MSA 4.05 (*Dieringer & Schlötterer, 2003*). The observed ($H_o$) and unbiased expected heterozygosity ($uH_E$), inbreeding coefficient ($F_{IS}$) as well as the number of deviations from Hardy-Weinberg expectations (HWE) were computed per sample using GenAlEx v6.1 (*Peakall & Smouse, 2006*). Linkage disequilibrium (LD) between pairwise loci per sample was computed using the program Genepop (*Rousset, 2008*). The false discovery rate (FDR) correction of *Benjamini & Hochberg (1995)* was applied to *p*-values to control for Type I errors.

In marine fish species, loci carrying signatures of locally divergent selection might function as powerful markers to evaluate spatially explicit genetic structure, as well as to

outline fisheries stocks for sustainable management (*Russello et al., 2012*). Thus, two approaches were combined to identify loci eventually departing from neutrality: BayeScan 2.1 (*Foll & Gaggiotti, 2008*) and Arlequin v.3.5.1.2 (*Excoffier, Laval & Schneider, 2005*). In BayeScan, sample size was set to 10,000 and thinning interval to 50; and loci with a posterior probability over 0.99, corresponding to a Bayes Factor >2 (*i.e.*, "decisive selection" (*Foll & Gaggiotti, 2006*), were retained as outliers. In Arlequin, analysis was simulated based on 1,000 demes with 50,000 simulations under a hierarchal island model.

Pairwise $F_{ST}$ (*Weir & Cockerham, 1984*) was computed using GENEPOP 4.0.6 (*Rousset, 2008*), whereas population differentiation was tested *via* G exact tests calculated using the following Monte Carlo Markov Chain (MCMC) parameters: 10,000 steps of dememorisation, and 5,000 iterations for 100 batches, also with GENEPOP 4.0.6.

The Bayesian clustering approach implemented in STRUCTURE v.2.3.4 (*Pritchard, Stephens & Donnelly, 2000*), and conducted using the software ParallelStructure (*Besnier & Glover, 2013*), was used to identify genetic groups under a model assuming admixture and correlated allele frequencies, both using and without using geographic LOCPRIORS to assist the clustering. Ten runs with a burn-in period consisting of 100,000 replications and a run length of 1,000,000 MCMC iterations were performed for K = 1 to K = 10 clusters. To determine the number of genetic groups, STRUCTURE output was analysed using two approaches: (a) the *ad hoc* summary statistic ΔK of *Evanno, Regnaut & Goudet (2005)* and (b) the *Puechmaille (2016)* four statistics (MedMedK, MedMeanK, MaxMedK and MaxMeanK), specially recommended for uneven sampling sizes, both conducted using StructureSelector (*Li & Liu, 2018*). Finally, the ten runs for the selected Ks were averaged with CLUMPP v.1.1.1 (*Jakobsson & Rosenberg, 2007*) using the FullSearch algorithm and the G′ pairwise matrix similarity statistic, and graphically displayed using barplots. Furthermore, the relationship among samples was also examined using the Discriminant Analysis of Principal Components (DAPC) (*Jombart, Devillard & Balloux, 2010*) implemented in the R (*R Core Team, 2020*) package *adegenet* (*Jombart, 2008*) in which groups were defined *a priori* using geographically explicit samples. To avoid overfitting, both the optimal number of principal components and discriminant functions to be retained were determined through cross validation using the *xvalDapc* function from adegenet (*Jombart & Collins, 2015*; *Miller, Cullingham & Peery, 2020*).

The relationship between genetic ($F_{ST}$) and geographic (km) distance was examined to investigate if it followed the expectations of an "Isolation by Distance" pattern (IBD), *i.e.*, increasing genetic differentiation with geographic distance as a result of restricted gene flow and drift (*Rousset, 1997*; *Slatkin, 1993*; *Wright, 1943*). A two-tailed *Mantel (1967)* test was conducted using PASSaGE v2 (*Rosenberg & Anderson, 2011*) and significance was assessed *via* 10,000 permutations. The matrix of pairwise shortest distance by water was created by calculating least-cost distances *via* seas (avoiding landmasses) between sampling sites using the *lc.dist* function from the R (*R Core Team, 2020*) package *marmap* v1.0 (*Pante & Simon-Bouhet, 2013*).

# RESULTS

## Quality control

Some 679 individuals out of the 1,848 genotyped ones were discarded due to exceeding the threshold of acceptance of missing genotypic data, which was set at 25% or due to unreliable scoring. From the 1,169 retained individuals, no missing markers were reported for 84% of the fish, whereas only 4% of the individuals displayed the maximum missing data allowed (see distribution of missing data per sample and locus in Fig. S1 in Supporting Information).

Data validation conducted with Micro-Checker did not find evidence of scoring error due to stuttering or large allelic dropout but suggested the possibility of null alleles in two of the loci, which showed a significant deficit of heterozygotes ($p < 0.0001$) after global Hardy-Weinberg tests. The frequency of null alleles per sample in these loci took a maximum value of 0.216 (locus STJ5 in ANG_2), whereas it was <0.05 in 71% of the cases (see Table S2 in Supporting information). As global and pairwise divergence calculated with and without using the ENA correction implemented in FreeNA displayed similar values (*i.e.*, a difference of 0.00045 in the overall $F_{ST}$ values); the two loci suggestive of displaying null alleles were retained for further analyses.

The simulation analyses revealed that the dataset carried enough statistical power to detect genetic differentiation as low as $F_{ST} = 0.0066$ in 80% of the cases, whereas only in 11% of the cases when levels of differentiation were five-fold lower ($F_{ST} = 0.0012$). The probability of rejecting the null hypothesis of panmixia if true (type I error) was estimated at 5% (Fisher test) and thus considered appropriate (Fig. S2 in Supporting Information). In addition, the number of microsatellite markers with the capacity of differentiating unique individuals can be inferred from the plateau of the genotype accumulation curve and this was achieved with some 37% of the markers used (Fig. S3 in Supporting Information).

## Summary statistics

The eight loci showed high rates of amplification success (93–100%) and suitable levels of polymorphism for population genetic analysis (100% polymorphic loci per sample). Allelic richness based on a minimum sample size of 20 diploid individuals took similar values throughout all the geographic range and moved between 11.5 (IVCO) and 13.6 (GAB_1), whereas the number of alleles for the same set of samples ranged between 96 and 137, respectively (Table 1). A total of 17 private alleles were found; they were distributed across all loci but SJT122 and ranged from 1 (Sco2-1, SJ78) to 4 (SJNT19, SJT199 and SJT53) and across 12 samples ranging from 1 (MOR_6, MOR_7, MOR_9, MAU_4, GAM, SIER, IVCO,GHA_1, GAB_1) to 4 (SEN_1). $H_o$ and $uH_E$ took similar values across samples and in the range 0.73–0.9. $uH_E$ was consistently higher than $H_O$ in all samples; in consequence, multilocus test detected heterozygote deficit in every sample but SIER ($p$-val = 0.257). The three loci displaying deviations from HWE in a larger number of samples after correcting for FDR were: SJT5 (in 20 samples), and Sco2-1 and SJ78 (in six). Both SJT5 and Sco2-1 exhibited homozygote excess and likely presence of null alleles.

## Genetic differentiation

No locus was found to be deviate from neutral expectations according to Arlequin, whereas BayeScan flagged all of them as putative candidates to balancing selection (Table S3 in Supporting Information). Due to the lack of consensus of neutrality between methods, the full set of microsatellites was retained for subsequent analyses.

Overall, no significant population structuring was detected across the study area as indicated by a global $F_{ST}$ of 0.001 ($p$ = 0.587). Pairwise $F_{ST}$ ranged from 0.000 to 0.011 (Table 2) with no significant differentiation whatsoever across the investigated geographic range.

STRUCTURE conducted both with and without LOCPRIORS reported a decreasing trend of LnP(D) across subsequent values of K with the highest average likelihood at K = 1 (Figs. S4A and S4B in Supporting Information). Evanno test, which by definition cannot yield K = 1, reported K = 2 (no priors) and K = 5 (LOCPRIORS), respectively but with extremely little support as both ΔK values were low (ΔK < 7). *Puechmaille (2016)*'s four statistics (MedMedK, MedMeanK, MaxMedK and MaxMeanK) implemented in StructureSelector confirmed one single genetic group in the model using priors, and K = 2 when not using them (Figs. S4A and S4B in Supporting Information). The barplot for K = 2 for the model without using LOCPRIORS revealed inferred ancestry to cluster ranging around 0.5 for each individual (Fig. S5 in Supporting Information), which is consistent with one single genetic unit.

Cross validation determined that 200 was the optimal number of PCs to be retained for the DAPC analysis using the 33 geographically explicit locations. In agreement with STRUCTURE, the corresponding DAPC plot revealed a large overlap of the individuals sampled across all the studied geographic range (Fig. 2) with none of the axes reaching 7% of the variation. Although some some individuals of SEN_1 partially deviated from the cloud, SEN_1 was not significantly different from any of the remaining samples according to pairwise $F_{ST}$.

Mantel test revealed lack of correlation between genetic differentiation (pairwise $F_{ST}$) and geographic distance along the latitudinal gradient ranging from Morocco to Namibia ($r_{xy}$ = 0.012, $p$ = 0.401).

# DISCUSSION

Investigating transboundary fish stocks with genetics is crucial for sustainable fisheries management. By identifying population structure in important commercial fish species, we can prevent overexploitation and allow for coordinated management strategies across national borders. This is the first attempt to describe the population structure of the Atlantic chub mackerel over most of its African distribution. The main result of the present study is the lack of geographically explicit structure in the coastline stretching from Morocco (27.39°N) to Namibia (22.21°S) based upon eight microsatellite markers.

## Genetic diversity

Many marine fishes exhibit high levels of genetic diversity due to historically large effective population sizes (*Hauser et al., 2002*; *Hauser & Carvalho, 2008*). All samples analysed in

**Table 2** Tests of genetic differentiation of *S. colias* samples collected from Northeast Atlantic waters of West Africa between Morocco and Namibia. Pairwise $F_{ST}$ values (bottom diagonal) and significance of G exact tests (upper diagonal).

| | MOR_1 | MOR_2 | MOR_3 | MOR_4 | MOR_5 | MOR_6 | MOR_7 | MOR_8 | MOR_9 | MAU_1 | MAU_2 | MAU_3 | MAU_4 | MAU_5 | MAU_6 | SEN_1 | SEN_2 | GAM | GUI_1 | GUI_2 | LIB | IVCO | GHA_1 | GHA_2 | GHA_3 | GHA_4 | GAB_1 | GAB_2 | ANG_1 | ANG_2 | NAM |
|---|---|---|---|---|---|---|---|---|---|---|---|---|---|---|---|---|---|---|---|---|---|---|---|---|---|---|---|---|---|---|---|
| MOR_1 | | 1.000 | 1.000 | 0.468 | 0.454 | 1.000 | 0.538 | 1.000 | 1.000 | 1.000 | 1.000 | 1.000 | 1.000 | 1.000 | 1.000 | 1.000 | 1.000 | 1.000 | 1.000 | 1.000 | 1.000 | 0.574 | 1.000 | 1.000 | 1.000 | 1.000 | 1.000 | 1.000 | 1.000 | 1.000 | 1.000 |
| MOR_2 | 0.000 | | 1.000 | 0.504 | 0.504 | 1.000 | 0.490 | 1.000 | 1.000 | 1.000 | 1.000 | 1.000 | 1.000 | 1.000 | 1.000 | 1.000 | 1.000 | 1.000 | 1.000 | 1.000 | 1.000 | 0.599 | 1.000 | 1.000 | 1.000 | 1.000 | 1.000 | 1.000 | 1.000 | 1.000 | 1.000 |
| MOR_3 | 0.000 | 0.000 | | 0.512 | 0.490 | 1.000 | 0.498 | 1.000 | 1.000 | 1.000 | 1.000 | 1.000 | 1.000 | 1.000 | 1.000 | 1.000 | 1.000 | 1.000 | 1.000 | 1.000 | 1.000 | 0.550 | 1.000 | 1.000 | 1.000 | 1.000 | 1.000 | 1.000 | 1.000 | 1.000 | 1.000 |
| MOR_4 | 0.000 | 0.001 | 0.000 | | 0.244 | 0.489 | 0.264 | 0.483 | 0.493 | 0.458 | 0.501 | 0.516 | 0.524 | 0.521 | 0.514 | 0.526 | 0.493 | 0.524 | 0.486 | 0.504 | 0.499 | 0.281 | 0.472 | 0.495 | 0.498 | 0.515 | 0.500 | 0.515 | 0.518 | 0.488 | 0.529 |
| MOR_5 | 0.000 | 0.001 | 0.000 | 0.000 | | 0.489 | 0.246 | 0.493 | 0.507 | 0.503 | 0.509 | 0.488 | 0.506 | 0.499 | 0.532 | 0.480 | 0.496 | 0.559 | 0.502 | 0.537 | 0.514 | 0.301 | 0.493 | 0.504 | 0.529 | 0.490 | 0.451 | 0.551 | 0.527 | 0.471 | 0.521 |
| MOR_6 | 0.000 | 0.000 | 0.000 | 0.000 | 0.000 | | 0.556 | 1.000 | 1.000 | 1.000 | 1.000 | 1.000 | 1.000 | 1.000 | 1.000 | 1.000 | 1.000 | 1.000 | 1.000 | 1.000 | 1.000 | 0.546 | 1.000 | 1.000 | 1.000 | 1.000 | 1.000 | 1.000 | 1.000 | 1.000 | 1.000 |
| MOR_7 | 0.000 | 0.000 | 0.002 | 0.000 | 0.000 | 0.002 | | 0.482 | 0.477 | 0.496 | 0.513 | 0.523 | 0.500 | 0.487 | 0.559 | 0.526 | 0.479 | 0.589 | 0.515 | 0.548 | 0.543 | 0.326 | 0.498 | 0.503 | 0.544 | 0.482 | 0.503 | 0.573 | 0.496 | 0.507 | 0.593 |
| MOR_8 | 0.000 | 0.001 | 0.000 | 0.000 | 0.001 | 0.001 | 0.002 | | 1.000 | 1.000 | 1.000 | 1.000 | 1.000 | 1.000 | 1.000 | 1.000 | 1.000 | 1.000 | 1.000 | 1.000 | 1.000 | 0.526 | 1.000 | 1.000 | 1.000 | 1.000 | 1.000 | 1.000 | 1.000 | 1.000 | 1.000 |
| MOR_9 | 0.000 | 0.000 | 0.001 | 0.000 | 0.000 | 0.000 | 0.000 | 0.000 | | 1.000 | 1.000 | 1.000 | 1.000 | 1.000 | 1.000 | 1.000 | 1.000 | 1.000 | 1.000 | 1.000 | 1.000 | 0.510 | 1.000 | 1.000 | 1.000 | 1.000 | 1.000 | 1.000 | 1.000 | 1.000 | 1.000 |
| MAU_1 | 0.000 | 0.000 | 0.000 | 0.002 | 0.000 | 0.001 | 0.004 | 0.002 | 0.001 | | 1.000 | 1.000 | 1.000 | 1.000 | 1.000 | 1.000 | 1.000 | 1.000 | 1.000 | 1.000 | 1.000 | 0.514 | 1.000 | 1.000 | 1.000 | 1.000 | 1.000 | 1.000 | 1.000 | 1.000 | 1.000 |
| MAU_2 | 0.000 | 0.003 | 0.002 | 0.002 | 0.001 | 0.000 | 0.000 | 0.000 | 0.000 | 0.002 | | 1.000 | 1.000 | 1.000 | 1.000 | 1.000 | 1.000 | 1.000 | 1.000 | 1.000 | 1.000 | 0.540 | 1.000 | 1.000 | 1.000 | 1.000 | 1.000 | 1.000 | 1.000 | 1.000 | 1.000 |
| MAU_3 | 0.000 | 0.000 | 0.000 | 0.000 | 0.000 | 0.001 | 0.000 | 0.000 | 0.000 | 0.000 | 0.000 | | 1.000 | 1.000 | 1.000 | 1.000 | 1.000 | 1.000 | 1.000 | 1.000 | 1.000 | 0.493 | 1.000 | 1.000 | 1.000 | 1.000 | 1.000 | 1.000 | 1.000 | 1.000 | 1.000 |
| MAU_4 | 0.000 | 0.000 | 0.000 | 0.001 | 0.003 | 0.000 | 0.004 | 0.000 | 0.000 | 0.002 | 0.002 | 0.000 | | 1.000 | 1.000 | 1.000 | 1.000 | 1.000 | 1.000 | 1.000 | 1.000 | 0.540 | 1.000 | 1.000 | 1.000 | 1.000 | 1.000 | 1.000 | 1.000 | 1.000 | 1.000 |
| MAU_5 | 0.000 | 0.000 | 0.000 | 0.005 | 0.006 | 0.004 | 0.005 | 0.010 | 0.007 | 0.005 | 0.006 | 0.004 | 0.008 | | 1.000 | 1.000 | 1.000 | 1.000 | 1.000 | 1.000 | 1.000 | 0.551 | 1.000 | 1.000 | 1.000 | 1.000 | 1.000 | 1.000 | 1.000 | 1.000 | 1.000 |
| MAU_6 | 0.002 | 0.001 | 0.008 | 0.005 | 0.006 | 0.004 | 0.003 | 0.002 | 0.002 | 0.000 | 0.006 | 0.004 | 0.005 | 0.002 | | 1.000 | 1.000 | 1.000 | 1.000 | 1.000 | 1.000 | 0.484 | 1.000 | 1.000 | 1.000 | 1.000 | 1.000 | 1.000 | 1.000 | 1.000 | 1.000 |
| SEN_1 | 0.000 | 0.000 | 0.001 | 0.001 | 0.004 | 0.002 | 0.003 | 0.002 | 0.002 | 0.000 | 0.004 | 0.000 | 0.002 | 0.001 | 0.001 | | 1.000 | 1.000 | 1.000 | 1.000 | 1.000 | 0.570 | 1.000 | 1.000 | 1.000 | 1.000 | 1.000 | 1.000 | 1.000 | 1.000 | 1.000 |
| SEN_2 | 0.000 | 0.000 | 0.000 | 0.000 | 0.000 | 0.000 | 0.000 | 0.000 | 0.000 | 0.001 | 0.002 | 0.000 | 0.000 | 0.001 | 0.005 | 0.001 | | 1.000 | 1.000 | 1.000 | 1.000 | 0.548 | 1.000 | 1.000 | 1.000 | 1.000 | 1.000 | 1.000 | 1.000 | 1.000 | 1.000 |
| GAM | 0.000 | 0.000 | 0.000 | 0.000 | 0.000 | 0.000 | 0.000 | 0.000 | 0.002 | 0.000 | 0.001 | 0.000 | 0.000 | 0.000 | 0.000 | 0.000 | 0.000 | | 1.000 | 1.000 | 1.000 | 0.499 | 1.000 | 1.000 | 1.000 | 1.000 | 1.000 | 1.000 | 1.000 | 1.000 | 1.000 |
| GUI_1 | 0.000 | 0.001 | 0.001 | 0.000 | 0.000 | 0.000 | 0.008 | 0.001 | 0.002 | 0.001 | 0.003 | 0.001 | 0.000 | 0.000 | 0.010 | 0.006 | 0.000 | 0.000 | | 1.000 | 1.000 | 0.496 | 1.000 | 1.000 | 1.000 | 1.000 | 1.000 | 1.000 | 1.000 | 1.000 | 1.000 |
| GUI_2 | 0.000 | 0.000 | 0.000 | 0.000 | 0.000 | 0.000 | 0.000 | 0.000 | 0.002 | 0.000 | 0.003 | 0.003 | 0.000 | 0.003 | 0.005 | 0.003 | 0.000 | 0.000 | 0.010 | | 1.000 | 0.508 | 1.000 | 1.000 | 1.000 | 1.000 | 1.000 | 1.000 | 1.000 | 1.000 | 1.000 |
| LIB | 0.000 | 0.003 | 0.001 | 0.001 | 0.002 | 0.003 | 0.004 | 0.001 | 0.003 | 0.002 | 0.004 | 0.000 | 0.005 | 0.004 | 0.010 | 0.007 | 0.000 | 0.001 | 0.006 | 0.008 | | 0.521 | 1.000 | 1.000 | 1.000 | 1.000 | 1.000 | 1.000 | 1.000 | 1.000 | 1.000 |
| IVCO | 0.000 | 0.001 | 0.004 | 0.000 | 0.000 | 0.001 | 0.002 | 0.000 | 0.000 | 0.000 | 0.000 | 0.000 | 0.000 | 0.000 | 0.004 | 0.002 | 0.000 | 0.000 | 0.000 | 0.004 | 0.005 | | 0.591 | 0.531 | 0.478 | 0.510 | 0.538 | 0.501 | 0.483 | 0.496 | 0.499 |
| GHA_1 | 0.000 | 0.000 | 0.001 | 0.000 | 0.002 | 0.006 | 0.000 | 0.000 | 0.002 | 0.000 | 0.003 | 0.003 | 0.001 | 0.003 | 0.005 | 0.000 | 0.002 | 0.003 | 0.004 | 0.004 | 0.005 | 0.003 | | 1.000 | 1.000 | 1.000 | 1.000 | 1.000 | 1.000 | 1.000 | 1.000 |
| GHA_2 | 0.000 | 0.000 | 0.000 | 0.000 | 0.000 | 0.000 | 0.000 | 0.000 | 0.000 | 0.000 | 0.000 | 0.000 | 0.000 | 0.000 | 0.004 | 0.000 | 0.000 | 0.000 | 0.002 | 0.002 | 0.002 | 0.000 | 0.000 | | 1.000 | 1.000 | 1.000 | 1.000 | 1.000 | 1.000 | 1.000 |

(Continued)

| | MOR_1 | MOR_2 | MOR_3 | MOR_4 | MOR_5 | MOR_6 | MOR_7 | MOR_8 | MOR_9 | MAU_1 | MAU_2 | MAU_3 | MAU_4 | MAU_5 | MAU_6 | SEN_1 | SEN_2 | GAM | GUI_1 | GUI_2 | LIB | IVCO | GHA_1 | GHA_2 | GHA_3 | GHA_4 | GAB_1 | GAB_2 | ANG_1 | ANG_2 | NAM |
|---|---|---|---|---|---|---|---|---|---|---|---|---|---|---|---|---|---|---|---|---|---|---|---|---|---|---|---|---|---|---|---|
| GHA_3 | 0.000 | 0.000 | 0.004 | 0.001 | 0.000 | 0.001 | 0.000 | 0.004 | 0.003 | 0.002 | 0.004 | 0.003 | 0.001 | 0.002 | 0.000 | 0.000 | 0.000 | 0.000 | 0.000 | 0.002 | 0.005 | 0.002 | 0.002 | 0.001 | | 1.000 | 1.000 | 1.000 | 1.000 | 1.000 | 1.000 |
| GHA_4 | 0.000 | 0.001 | 0.000 | 0.001 | 0.000 | 0.000 | 0.004 | 0.002 | 0.002 | 0.002 | 0.004 | 0.002 | 0.000 | 0.002 | 0.009 | 0.004 | 0.003 | 0.000 | 0.003 | 0.006 | 0.002 | 0.004 | 0.001 | 0.000 | 0.005 | | 1.000 | 1.000 | 1.000 | 1.000 | 1.000 |
| GAB_1 | 0.000 | 0.001 | 0.001 | 0.001 | 0.001 | 0.001 | 0.000 | 0.001 | 0.002 | 0.005 | 0.001 | 0.003 | 0.000 | 0.004 | 0.008 | 0.005 | 0.000 | 0.001 | 0.001 | 0.008 | 0.004 | 0.000 | 0.000 | 0.001 | 0.002 | 0.002 | | 1.000 | 1.000 | 1.000 | 1.000 |
| GAB_2 | 0.000 | 0.000 | 0.003 | 0.000 | 0.000 | 0.000 | 0.000 | 0.000 | 0.001 | 0.003 | 0.000 | 0.006 | 0.000 | 0.003 | 0.008 | 0.002 | 0.000 | 0.000 | 0.001 | 0.004 | 0.005 | 0.000 | 0.000 | 0.000 | 0.000 | 0.005 | 0.000 | | 1.000 | 1.000 | 1.000 |
| ANG_1 | 0.000 | 0.000 | 0.000 | 0.000 | 0.000 | 0.000 | 0.000 | 0.001 | 0.000 | 0.000 | 0.000 | 0.000 | 0.000 | 0.001 | 0.006 | 0.003 | 0.000 | 0.000 | 0.000 | 0.008 | 0.006 | 0.000 | 0.000 | 0.000 | 0.002 | 0.000 | 0.000 | 0.000 | | 1.000 | 1.000 |
| ANG_2 | 0.000 | 0.001 | 0.004 | 0.006 | 0.003 | 0.002 | 0.003 | 0.001 | 0.003 | 0.004 | 0.000 | 0.004 | 0.000 | 0.001 | 0.008 | 0.003 | 0.002 | 0.000 | 0.001 | 0.003 | 0.010 | 0.000 | 0.001 | 0.000 | 0.001 | 0.002 | 0.000 | 0.000 | 0.000 | | 1.000 |
| NAM | 0.000 | 0.000 | 0.000 | 0.001 | 0.002 | 0.001 | 0.000 | 0.000 | 0.000 | 0.001 | 0.001 | 0.000 | 0.000 | 0.000 | 0.006 | 0.001 | 0.000 | 0.000 | 0.000 | 0.011 | 0.003 | 0.001 | 0.003 | 0.000 | 0.001 | 0.004 | 0.000 | 0.000 | 0.000 | 0.000 | |

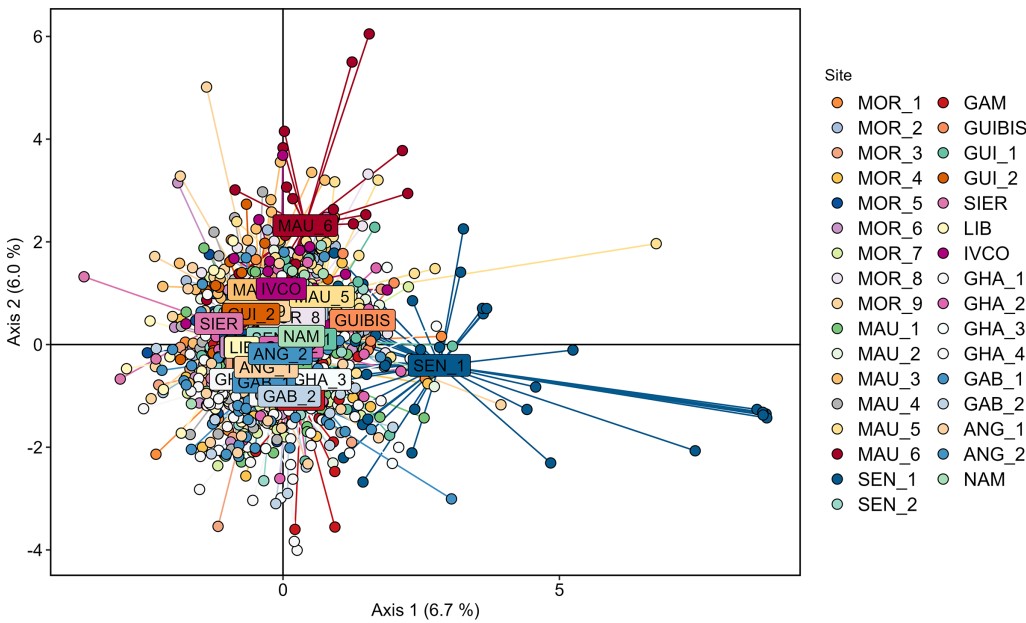

**Figure 2 Genetic differentiation among *Scomber colias* sampled in the West African coast and genotyped at eight microsatellite markers depicted using Discriminant Analysis of Principal Components (DAPC) after retaining 200 principal components.** Individuals from different samples are represented by coloured dots, and name labels are placed on the centroids of each geographically-explicit site.

the current study showed high levels of genetic diversity in terms of allelic richness (11.5 to 13.6) and heterozygosity (≥0.7) in agreement with results obtained for other medium and small pelagic species, *e.g.*, *S. australasicus* in the western North Pacific (*Tzeng et al., 2009*), European sprat *Sprattus sprattus* (*Glover et al., 2011*), European anchovy *Engraulis encrasicolus* (*Zarraonaindia et al., 2009*) or European pilchard *Sardina pilchardus* (*Baibai et al., 2012*) in the North-East Atlantic. Overall, all samples (with the exception of the one from Sierra Leone represented by just eleven individuals) deviated from HWE expectations showing a deficit of heterozygotes. Heterozygote deficiency is not new to *Scomber* literature as it has been shown in microsatellite studies of *S. colias* in the East Atlantic Ocean and the Mediterranean Sea (*Medina-Alcaraz, 2016*) as well as in *S. japonicus* along the Chinese coast (*Cheng, Zhu & Chen, 2014*; *Zeng, Cheng & Chen, 2012*). Heterozygote deficiency in non-selfing, diploid populations is relatively common (*Waldman & McKinnon, 1993*) and in largely outcrossing species, it has commonly been interpreted as indirect evidence of the mixing of differentiated gene pools (Wahlund effect). However, selection and the presence of null alleles are common cause of heterozygote deficiency. In agreement with the latter, in the current study, two out of the eight genotyped loci were likely to display null alleles according to Micro-Checker assessment. Finally, and given the *S. colias* stocks are considered to be overexploited along West Africa (*FAO, 2020*), the third hypothesis to explain heterozygote deficiency could be overexploitation as overfishing reduces population size drastically, leading to genetic bottlenecks and loss of alleles (*Smith, 1994*).

## Genetic population structure

In this study, the lack of genetic differentiation in *S. colias* revealed by eight microsatellites in the geographic area from Morocco (27.39°N) to Namibia (22.21°S) supporting a theory of panmixia would be a foreseen outcome due to pertinent ecological features of pelagic fish, such as high fecundity, large effective population sizes that impose limitations to genetic drift as well as a notable dispersal capacity as planktonic larvae and free-swimming adults that promotes gene flow (*e.g.*, *Carlsson et al., 2004*; *Martínez et al., 2006*; *Waples, 1987, 1998*). The combination of both features acts as a major homogenizing force hindering genetic differentiation and eventually leading to panmixia (*Canales-Aguirre et al., 2018*; *De Bie et al., 2012*; *Roy et al., 2014*; *Weersing & Toonen, 2009*).

*S. colias* is a highly migratory species with individuals capable of covering extensive distances during their annual migrations favoured by a body shape well fitted for swift motion and free swimming mode of life (*Collette, 1999*; *Uriarte et al., 2001*). In addition, its planktonic larval stages also display considerable dispersal capabilities as pelagic larvae can be transported over vast distances by ocean currents and eddies (*Hernández-León, Gómez & Arístegui, 2007*) effectively leading to genetic homogeneity among populations. In agreement, other studies have revealed a notable absence of genetic differentiation on *S. colias* in the Northeast Atlantic and adjacent waters suggesting a large panmictic unit (*Rodríguez-Ezpeleta et al., 2016*; *Stroganov et al., 2023*; *Zardoya et al., 2004*). Likewise, congeneric species such as *S. scombrus* reveal population differentiation only at a transatlantic scale (*Nesbø et al., 2000*), whereas *S. japonicus* displays weak genetic structure across the NW Pacific (*Cheng et al., 2015*).

The hydrological conditions along the West African coast play a crucial role in various ecological processes and are influenced by factors such as ocean currents, eddies, temperature gradients, and upwelling phenomena. The region experiences the influence of several major ocean currents (see Fig. 1), including the Canary Current, the Guinea Current, and the Benguela Current, which affect the distribution of nutrients and plankton, which in turn impacts the abundance and distribution of marine life, including fish populations and seem to shape genetic differentiation of coastal species in this region (*Alpers et al., 2013*; *Barton, Field & Roy, 2013*; *Mittelstaedt, 1991*; *Nielsen et al., 2018*; *Pelegrí & Peña-Izquierdo, 2015*; *Shannon, 2001*). In consequence, distant sites may be connected by a strong current between them, so, the absence of the genetic structure that can result from the long-distance dispersal of the early development stages (*Vásquez et al., 2013*; *White et al., 2010*).

However, regional differences in various life-history traits with latitudinal trends have been detected in both European and NW African waters, with the Strait of Gibraltar as changing point (*ICES, 2021*). Several studies have addressed the delineation of *S. colias* population structure in the East Atlantic using phenotypic characters, and some latitudinal trends seem to occur in length composition and life-history traits from the Strait of Gibraltar northwards into European waters and southwards, through African waters (*Domínguez-Petit et al., 2022*; *ICES, 2021*; *Landa et al., 2022*; *Perrotta, Carvalho & Isidro, 2005*). Differences include size at first maturity, which seems to be larger in the South of Africa than in the North (*ICES, 2021*). Likewise, the spawning period, a crucial

evolutionary driver in pelagic species (*e.g.*, *Lamichhaney et al., 2017*; *Petrou et al., 2021*), corresponds to January-March in the region comprising Morocco to Senegal, in contrast to June-September in the area from Ghana to South Africa (*ICES, 2021*). Regarding the Atlantic coast of North America, the population structure of the Atlantic chub mackerel has not been extensively studied; however, differences in growth (*Daley & Leaf, 2019*), habitat suitability (*Chen et al., 2009*), spawning seasons (*Weber & McClatchie, 2011*), size at maturity (*Cerna & Plaza, 2014*) and morphology (*Erguden et al., 2009*) suggest that different sub-stocks might exist. The latitudinal trends observed in the East Atlantic, with the Strait of Gibraltar as inflection point (*ICES, 2021*), indicate that significant differences might be expected if individuals from the whole East Atlantic distribution of *S. colias* (including Mediterranean samples) were jointly analysed. This study is lacking samples from North of Morocco as well as from the Iberian Peninsula, which should be included in future genetic assessments.

## Management implications

The lack of geographically explicit structure found here suggests the existence of one single putative genetic stock for *S. colias* inhabiting the West African waters from Morocco (27.39°N) to Namibia (22.21°S). Panmixia in a transboundary stock can present both opportunities and challenges. First, if a transboundary stock truly acts as a single panmictic population, then a single stock assessment might be sufficient, and the complexity of modelling population dynamics and setting catch quotas becomes simplified. In this case, all countries sharing the stock would benefit from implementing consistent management practices and should work together to set catch quotas and implement regulations that benefit the entire population.

However, a determination of panmixia based on a limited dataset might mask the presence of subtle subpopulations with distinct spawning grounds or migration patterns. In this case, management might need to consider specific population components, which if unnoticed could lead to inadequate management strategies and potential overexploitation of vulnerable subpopulations.

Although the sampling strategy of the present study comprises most of the West African geographical distribution of the species, this did not include the entire NE Atlantic distribution and the seasonal migrations could not be considered in the study. Therefore, although genetic divergences are not noticeable, some kind of local adaptations seem to exist, more when seasonal migration processes of the species have been described in the study area (*García, 1982*). From this perspective, and considering the current global warming scenario in which the northwards expansion of the species across the East Atlantic Ocean from regions of higher abundance off northwest Africa to the waters of the Atlantic Iberian and the Mediterranean Sea is a fact (*Jurado-Ruzafa et al., 2024*), fisheries management of Atlantic chub mackerel should account not only with punctual assessment statuses of political-based stocks but consider all the variables mentioned along the present section.

Although our data suggests that a nearly panmictic model is the most plausible scenario for the genetic population pattern of *S. colias*, implying high level of genetic connectivity

across the Western African countries ranging from 28°N to 22°S. However, phenotypic variability and life history of *S. colias*, which follows a latitudinal gradient, support the evidence of regional demographic units as suggested by *Sbiba et al. (2024)* based upon otolith shape. Therefore, relying solely on genetic data may not provide a complete snapshot of the demographic connectivity, especially in wild marine fish populations characterized by high migration rates and large population sizes. It is crucial to consider demographic connectivity in addition to genetic connectivity when managing an important commercial species to ensure effective and sustainable measures. Finally, it cannot be neglected that genetic differentiation could have eluded the scrutiny of the set of markers used here as well as the possible impact of ascertainment bias driven by markers isolated from sampled from the Pacific ocean (*Zeng & Cheng, 2012*). Future studies would benefit from the genomic tools currently available such as the genome assembly of this species recently published (*Machado et al., 2022*).

## CONCLUSIONS

The findings of this study shed light on the population genetics of *Scomber colias* along the West African coast. The study indicates that the species has very low genetic population differentiation and large migratory (dispersal) potential. These results have significant implications for the management of this crucial pelagic species in this region. The study emphasizes the need for cooperation efforts and further research to fully comprehend the underlying mechanisms of population connectivity and genetic homogenization. Advanced genomic tools, such as whole-genome sequencing in combination with phenotypic and environmental information could provide a more comprehensive understanding of any complex population structure or adaptive potential of *S. colias*. Such an understanding is important for developing more rational effective management strategies and ensuring the sustainability of this shared stock. It could also help refine our understanding of this species' evolutionary trajectories and guide sustainable management approaches at fine scale.

## ACKNOWLEDGEMENTS

We thank the captain and crew of R.V. "Dr. Fridtjof Nansen" for their invaluable assistance and contribution to this study, which made a significant impact on the research's success. Likewise, we thank Alejandro Mateos-Rivera for his advice in the laboratory. The authors would like to extend their gratitude to the INRH for nominating Salah eddine Sbiba to embark on the R/V Dr. Fridtjof Nansen. This opportunity has significantly contributed to the success of our research. Disclaimer: The views expressed in this publication are those of the author(s) and do not necessarily reflect the views or policies of the Food and Agriculture Organization of the United Nations.

### Funding

This study was funded by EAF-Nansen project, a collaboration between the Food and Agriculture Organisation of the United Nations, the Norwegian Agency for Development

Cooperation, and the Institute of Marine Research in Norway. This project also received support from Global Environmental Facility, as well as the Canary Current Large Marine Ecosystem project, implemented by the FAO, the United Nations Environment Program, and seven partner countries: Morocco, Mauritania, Senegal, The Gambia, Cape Verde, and Guinea Bissau. The funders had no role in study design, data collection and analysis, decision to publish, or preparation of the manuscript.

## Grant Disclosures

The following grant information was disclosed by the authors:
Food and Agriculture Organisation of the United Nations.
Norwegian Agency for Development Cooperation.
Institute of Marine Research in Norway.
Global Environmental Facility.
Canary Current Large Marine Ecosystem Project.

## Competing Interests

The authors declare that they have no competing interests.

## Author Contributions

- Salah eddine Sbiba conceived and designed the experiments, performed the experiments, analyzed the data, authored or reviewed drafts of the article, and approved the final draft.
- María Quintela analyzed the data, prepared figures and/or tables, authored or reviewed drafts of the article, and approved the final draft.
- Johanne Øyro performed the experiments, authored or reviewed drafts of the article, and approved the final draft.
- Geir Dahle performed the experiments, authored or reviewed drafts of the article, and approved the final draft.
- Alba Jurado-Ruzafa conceived and designed the experiments, authored or reviewed drafts of the article, and approved the final draft.
- Kashona Iita conceived and designed the experiments, authored or reviewed drafts of the article, and approved the final draft.
- Nikolaos Nikolioudakis conceived and designed the experiments, prepared figures and/or tables, authored or reviewed drafts of the article, and approved the final draft.
- Hocein Bazairi conceived and designed the experiments, authored or reviewed drafts of the article, and approved the final draft.
- Malika Chlaida conceived and designed the experiments, authored or reviewed drafts of the article, and approved the final draft.

## Field Study Permissions

The following information was supplied relating to field study approvals (*i.e.*, approving body and any reference numbers):
    The EAF-Nansen Programme approved the study (GCP/GLO/690/NOR).

## Data Availability

Genotype data is available at Zenodo and Havforskningsinstituttet:

Sbiba, S. E. (2024). Microsatellite data on *Scomber colias* in NW Africa [Data set]. Zenodo. https://doi.org/10.5281/zenodo.11518098

https://imr.brage.unit.no/imr-xmlui/handle/11250/3072605

## Supplemental Information

Supplemental information for this article can be found online at http://dx.doi.org/10.7717/peerj.17928#supplemental-information.

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
