# Peer review of "Genetic investigation of population structure in Atlantic chub mackerel, *Scomber colias* Gmelin, 1789 along the West African coast"

_PeerJ, doi:10.7717/peerj.17928_

## Round 0.1 · original submission · Major Revisions

The authors have written an interesting and well-structured manuscript on genetic structure in chub mackerel, and the results on haplotype structuring without geographic structure may be helpful in the management of this species. Is it possible the authors found a genomic inversion within this microsatellite allele that leads to this three-cluster result?

All three reviewers agree that this paper is of sufficient interest for the journal and have some minor reviews that I encourage the authors to take into account in their revisions.

A reviewer and I share a more significant concern that the allele scoring is flawed and over-estimating allelic diversity. It is important to genotype a subset of individuals more than once (as the reviewer suggests) to evaluate consistency among PCRs and allele calling, especially when not directly sequencing microsatellites with an Illumina or other platform. I suggest the authors review their microsatellite allele bins and if possible re-genotype a subset of individuals to provide increased confidence in the allelic diversity described in the paper.

Reviewer 1 ·

Basic reporting

The paper is well written with a clear professional English excluding minor grammatical issues.

Introduction and background are sufficient and comprehensive, and set out well the aims and relevance of the study.

Structure conforms to PeerJ standards except for:
a) Subheadings are not followed by a period and the corresponding text.
b) There is no Conclusion.

Figures are relevant, with good quality, well described and labelled.

Raw data was supplied and checked.

Experimental design

Original primary research within the Scope of the journal.

The research question was well defined, is relevant and meaningful. The Introduction provides supporting evidence showing the need and relevance of the study.

I have doubts on the rigor of the raw data collection as I detail bellow in 3. Validity of the Findings.

Methods are described with sufficient detail to allow replication, except for the allele scoring method which has raised strong doubts on its rigor and accuracy.

Validity of the findings

My major concern with the paper has to do with the allele scoring and genotyping approach used. One thing that I found striking was the very high allele diversity reported in the study (Table S1). That in itself would not be impossible but there are several observations reported in the paper that show that allele diversity may be strongly overestimated. For instance, alleles are listed for locus SJTN86 in Table S5, which is a perfect dinucleotide repeat microsatellite (Table S1), but the vast majority of the alleles differ in a single base pair only. Moreover, based on the type of repeat unit and size range of each marker shown in Table S1, one can also clearly see that the number of alleles is much larger than would be expected if most alleles would differ in size by the number of nucleotides in the repeat unit: locus SJ32 is an imperfect dinucleotide repeat marker with alleles differing in size by a maximum of 100 bp but for which 72 alleles (!) were scored, i.e. much more than the 50 that one would expect. The same rational applies (at least) to loci SJTN66, SJTN86, SJT216, Sco2_1, SJ78. Looking at graph from Figure S4, and graphs made with allele counts per locus based on the raw data made available by the authors, this apparent allele mis-scoring is made clearer: alleles of lower frequencies differ by 1-3bp from the most frequent ones (for dinucleotide and tetranucleotide repeat markers, respectively). E.g. in locus Sco2_1, a tetranucleotide repeat marker, has high frequency alleles of sizes 141, 145, 149, 153, 157, 161, 165, 169, etc. (which differ by 4 bp), and lower frequency alleles (likely “artificial”) at 144, 150, 151, 154, 158, 163, etc…

This observation may be due to the fact that the authors are classifying allele peaks in the chromatogram differing by a single base pair as different alleles. They mention in the Methods (lines 192-193) that alleles were automatically binned and then manually checked by to researchers. I would recommend that allele binning for dinucleotide repeat markers is made by the authors assuming 2 bp wide bins, and 4 bp wide bins for tetranucleotide repeat markers, as there may be variability in peak size between PCR reactions and sequencing runs that would lead to the same allele being shifted by a 1-3bp. I would also strongly recommend that a subset of the samples is run twice (e.g. 20% of the total) to assess allele scoring consistency among PCRs and sequencing runs.

As is, the likely overestimation of allele diversity may impact the results in two different ways:

1. The heterozygote deficiency reported for all markers in lines 317-318 may be due to individuals being genotyped as homozygotes for an allele that is incorrectly scored as distinct from the closest more frequent allele. Low frequency alleles, as those incorrectly scored as distinct, would be more likely to occur in heterozygosity in large populations at HWE; thus, homozigosity at these alleles would result in heterozygote defficiency as seen here.
2. Incorrectly scoring individuals as heterozygotes for the “artificial” alleles may also falsely increase observed heterozygosity. This will result in overall lower maximum Fst values as Fst is bound by the level of gene diversity (i.e. expected heterozygosity; see Meirmans, P.G. and Hedrick, P.W., 2011. Assessing population structure: FST and related measures. Molecular ecology resources, 11(1), pp.5-18.), and may bias the estimation of among-population genetic differentiation.

As such, the conclusions of the paper may be compromised by the issues associated with data collection.

Additional comments

The authors should rescore the alleles and re-do the genotyping of the samples following the recommendations made above, and re-run a subset of their samples to check for allele scoring consistency. Once the data are robust, then they can perform the sample data analyses as already outlined in the original manuscript and adjust their conclusions accordingly.

Reviewer 2 ·

Basic reporting

Sbiba et al. performed a population genetics investigation to define the genetic structure of Atlantic chub mackerel populations along the West African coast. The authors utilised 12 microsatellite markers to demonstrate that the examined populations exhibited genetic homogeneity. This suggests that there is a significant exchange of genes between populations, indicating the absence of any geographical or oceanic barriers to gene flow. The manuscript has exemplary writing and is highly compatible with the diverse readership of PeerJ. The reviewer has provided the authors with only a few minor revisions to consider:

Line 33: Write SSR in full. Although, the term switches to ‘microsatellite’ in Line 37 and ‘microsatellites’ as a keyword. In general, if a word appears in the title, it should not be included in the list of keywords. Please use microsatellite(s) throughout given its use in the title of the manuscript.

Line 136-138: This section is well written, but it could benefit from defining “microsatellite DNA” and its use in fisheries genetics given that the title contains the phrase “Microsatellite analyses.”

Experimental design

The reviewer identified no significant deficiencies in the analytical methodologies, nor any concerns pertaining to the data, interpretation of the findings, and subsequent discussion. The study’s sampling strategy is highly commendable.

Line 136-138: More information is needed here for how the runs including population information were conducted. Please clarify that you used the LOCPRIOR model and report on the model parameter ‘r’ so we can see if adding population information was any useful.

Validity of the findings

The supplied data is comprehensive, reliable, and has been carefully controlled to ensure statistical validity. The reported findings offer new and significant insights into the studied system, which have important implications for the management of fisheries.

Additional comments

no comment

Reviewer 3 ·

Basic reporting

Please see attached comments.

Experimental design

Please see attached comments.

Validity of the findings

The key finding of the article is that using microsatellites, the species under study does not exhibit quantifiable genetic differentiation within the sample range. The authors find a Wahlund effect and IBD, but they do not highlight findings based on the data presented. I would also like to see a more prominent acknowledgment of the limitations of the genetic tool (microsatellites) used in the discussion. Based on this comment, do you think the title is too equivocal? Maybe simply “Genetic investigation of population structure in Atlantic chub mackerel, Scomber colias Gmelin, 1789 along the West African coast”.

Additional comments

Please see attached comments

Annotated reviews are not available for download in order to protect the identity of reviewers who chose to remain anonymous.

---

## Round 0.2 · Major Revisions

Two reviewers were able to provide comments on this revised draft, and while both reviewers agree that several aspects of the manuscript have improved, additional revisions are needed. In particular, one reviewer (and myself) still have some concerns with the allele scoring. The authors do not currently provide adequate evidence of improved allele binning based on reviewer comments from the first draft, and also have not provided an updated dataset through their open data link for the reviewers to double check. I therefore ask that the authors follow the suggestions from reviewer one to provide confidence in the allele scoring, particularly where some loci showed large increases in allele numbers.

Reviewer 1 ·

Basic reporting

The paper is well written with a clear professional English excluding minor grammatical issues.

Introduction and background are sufficient and comprehensive, and set out well the aims and relevance of the study.

Figures are relevant, with good quality, well described and labelled.

Raw data was not supplied.

Experimental design

Original primary research within the Scope of the journal.

The research question was well defined, is relevant and meaningful. The Introduction provides supporting evidence showing the need and relevance of the study.

I have doubts on the rigor of the raw data collection as I detail bellow in 3. Validity of the Findings.

Validity of the findings

My main concern in the original version of the MS was the potential lack of accuracy and rigor in allele scoring. Specifically, based on the large number of alleles per locus and on Supplemental Figure S4 of the original MS, there was evidence that this was indeed one major issue in the data. The authors state they have found an error in the data but that it was due to issues in “parsing the genotypes and producing the final collated raw data”. Although they made the effort of repeating the PCR and allele scoring for 7% of the individuals (~25 out of 356), they made no attempt to change their allele scoring approach as they decided “the issues with the data collection seemed to be the main cause of the inconsistencies in allele scoring.”. While I sincerely wish that all issues with data quality were indeed solved, I have no way to confirm that this is indeed the case: unlike in the original version of the MS, there are no data similar to the one in the original Supplemental Figure S4 provided in the revised version, nor are there raw data available for inspection. Table S1 shows some changes in allele numbers for several loci, mostly towards reduction (4 loci), but locus SJNT86 had 20 originally scored alleles but increased to 51 (!) alleles in this revised version.

As such, my main issues with allele scoring and data quality remain.

Reviewer 2 ·

Basic reporting

The revised manuscript by Sbiba et al. is much improved following reviewer feedback. The manuscript requires minor editorial to correct some grammatical errors. Please see attached PDF file.

Experimental design

no comment

Validity of the findings

no comment

Annotated reviews are not available for download in order to protect the identity of reviewers who chose to remain anonymous.

---

## Round 0.3 · Minor Revisions

The authors have done an excellent job of addressing the major revisions from several reviewers and I believe that only minor revisions are now required. Reviewer 2 recommends making the objectives of the study very clear, as the goal of the study is still not completely apparent by the end of the introduction. There is also some information included in the discussion that should be mentioned in the introduction (and cited). Please ensure that all of the information, including reference volumes and page numbers etc. are included in your reference list as well (e.g., Carvalho et al. reference).

Reviewer 1 ·

Basic reporting

The paper is well written but improved written English is needed in the Discussion. I have made several suggestions on this topic throughout the MS.

Introduction and background are sufficient and comprehensive, and set out well the aims and relevance of the study.

Figures are relevant, with good quality, well described and labelled.

Raw data was supplied.

Reference list appears incomplete and/or lacking information of journal, volume and pages. I have found some of these issues twice while reading the MS and strongly advise the authors to check their citations in the text and the list of references.

Experimental design

Original primary research within the Scope of the journal.

The research question was well defined, is relevant and meaningful. The Introduction provides supporting evidence showing the need and relevance of the study.

Issues in the original paper regarding allele scoring and data collections have been corrected.

Validity of the findings

Upon inspection of the raw data and allele binning and frequency, I am convinced that the authors have addressed my previous concerns by re-scoring the microsatellite genotypes and filtering out both loci and individuals with ambiguous or low-quality data.

Conclusions are clear and linked to the original research question.

Reviewer 3 ·

Basic reporting

Significant improvements have been made between the various submissions by these authors. However, some significant changes in the introduction (missing citations, flow of information, study justification) and in the discussion (similar issues) are needed before this manuscript is suitable for publication.

Regarding the updates to the marker scoring and results, I do not have special expertise in the scoring of microsatellites and I defer to the other two reviewers for their expertise on this matter.

I agree with the justification to add an additional author.

Comments specific to manuscript sections :
Abstract:
See annotated document.

Introduction:
I feel the introduction is still lacking important information and context for your study. I get to the end of the introduction, and I am still not clear about the reason for the study or its objectives. To assist, I have provided suggestions to improve the introduction in an annotated word document provided.

In the discussion on line 383 you write – “In agreement, other studies have revealed a notable absence of genetic differentiation on S. colias in the Northeast Atlantic and adjacent waters suggesting a large panmictic unit (Rodríguez-Ezpeleta et al. 2016; Stroganov et al. 2023; Zardoya et al. 2004)”. These references and their findings need to be in your introduction.
Likewise, you cite Zeng & Cheng 2012 in your methods. The same authors have performed a study of a similar species S. japonicus in the Northwestern Pacific. They used microsatellites and found local genetic differentiation which they link to differences in spawning time and migratory behavior, yet this is not cited in the introduction despite being relevant.

Line 72-78 says: “The genus Scomber has been the object of several taxonomic revisions, and it is currently described as consisting of four species (Collette 1999): S. colias, S. japonicus Houttuyn, 1782, S. scombrus Linnaeus, 1758 and S. australasicus Cuvier, 1832. The taxonomic debate has been particularly intense regarding the discrimination between S. colias and S. japonicus, but the combined phylogenetic analysis of nuclear 5S rDNA sequences (Infante et al. 2007), together with significant mitochondrial DNA divergence (Catanese et al. 2010b; Trucco & Buratti 2017), decisively supported the distinction between S. japonicus distributed in the Indo-Pacific, and S. colias, in the Atlantic Ocean”. How is this relevant to the study in question? See the annotated word document for suggestions.
Line 91 – What’s a hinge zone? It’s not clear from the reference provided.
Some sentence structure and spelling needs addressing:
Line 65 - Change “tools” to tool.
Line 99 – Change to “Overall, these studies also highlight the high potential of phenotypic plasticity of this species”.

Methods:
See annotated file provided.
Regarding the updates to the marker scoring and results, I do not have special expertise in the scoring of microsatellites and I defer to the other two reviewers and their expertise on this matter.

Results:
See annotated file provided.

Table 2 – Change title to “Tests of genetic differentiation (FST) of S. colias samples collected from North East Atlantic waters of west Africa between Morocco and Namibia. Pairwise FST values (bottom diagonal) and significance of G exact tests (upper diagonal)”.

Discussion:
The discussion needs review and is missing important discussion on the limitations of this study to detect population structure.
Line 355 onward – This section on “Genetic population structure” does not mention the limitations of the genetic markers you have used. The majority of the markers you have used here were developed for S. japanoicas, and even though it has been shown to have successful cross-species amplification I am left wondering if you think this point may limit your ability to detect population level differentiation in another, separate, species? Also, many studies also now use SNPs. What could you potentially find if you have been able to use these markers?

Experimental design

See annotated document.

Validity of the findings

See annotated document.

Additional comments

See annotated document for my suggested changes and comments.

Annotated reviews are not available for download in order to protect the identity of reviewers who chose to remain anonymous.

---

## Round 0.4 · accepted · Accept

I thank the authors and reviewers for a thoughtful and comprehensive review process, which I believe improved this manuscript. The authors have conducted an excellent scientific study that I believe will be useful in management of this species, and provide background for future high-throughput sequencing approaches if warranted. I believe this manuscript is now ready for publication.